# iFormer: Integrating ConvNet and Transformer for Mobile Application

**Chuanyang Zheng**
Independent Researcher
chuanyang_zheng@sjtu.edu.cn

## Abstract

We present a new family of mobile hybrid vision networks, called iFormer, with a focus on optimizing latency and accuracy on mobile applications. iFormer effectively integrates the fast local representation capacity of convolution with the efficient global modeling ability of self-attention. The local interactions are derived from transforming a standard convolutional network, *i.e.*, ConvNeXt, to design a more lightweight mobile network. Our newly introduced mobile modulation attention removes memory-intensive operations in MHA and employs an efficient modulation mechanism to boost dynamic global representational capacity. We conduct comprehensive experiments demonstrating that iFormer outperforms existing lightweight networks across various tasks. Notably, iFormer achieves an impressive Top-1 accuracy of 80.4% on ImageNet-1k with a latency of only 1.10 ms on an iPhone 13, surpassing the recently proposed MobileNetV4 under similar latency constraints. Additionally, our method shows significant improvements in downstream tasks, including COCO object detection, instance segmentation, and ADE20k semantic segmentation, while still maintaining low latency on mobile devices for high-resolution inputs in these scenarios. Code and models are available at: https://github.com/ChuanyangZheng/iFormer.

## 1 Introduction

Building lightweight neural networks facilitates real-time analysis of images and videos captured by mobile applications such as smartphones. This not only enhances privacy protection and security by processing data locally on the device but also improves overall user experience.

Through the decades, convolutional neural networks (CNNs) (Krizhevsky et al., 2012; Szegedy et al., 2015; He et al., 2016) have emerged as the primary choice for balancing latency and performance on resource-constrained mobile devices. However, a significant limitation of CNNs is their reliance on a local sliding window mechanism, which imposes crucial inductive biases that may hinder modeling flexibility. Recently, the soaring development of vision transformers (ViTs) (Dosovitskiy et al., 2020) has begun to dominate various computer vision tasks, including image classification (Zhai et al., 2022), object detection (Liu et al., 2021a), and semantic segmentation (Xie et al., 2021). The core mechanism underlying ViTs is self-attention, which dynamically learns interactions between all image patches. This enables the model to focus on important regions adaptively and capture more global features. Nevertheless, deploying ViTs on mobile devices with limited resources poses significant challenges. On the one hand, the quadratic computational complexity of attention renders them unsuitable for large feature maps, which are common in the early stages of vi-

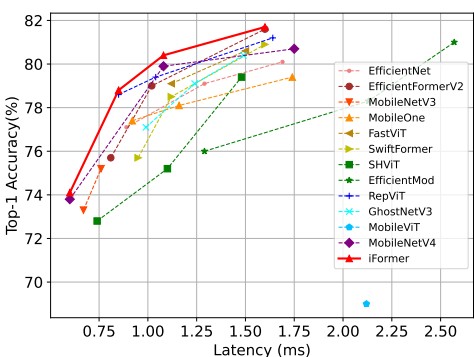

Figure 1: **Comparison of latency and accuracy between our iFormer and other existing methods on ImageNet-1k.** The latency is measured on an iPhone 13. Our iFormer is Pareto-optimal.

sion networks. On the other hand, the multi-head mechanism requires reshaping operations, leading to increased memory usage.

Many research efforts are devoted to combining the advantages of both CNNs and ViTs in designing lightweight networks while mitigating inefficient operations in mobile applications. Some studies (Zhang et al., 2023; Wang et al., 2024; Ma et al., 2024) revisit the architectural designs of lightweight CNNs from a ViT perspective and incorporate key components that contribute to the performance of ViTs into CNNs. Although these pure lightweight CNNs show improved performance compared to previous mobile networks (Howard et al., 2017; Zhang et al., 2018; Sandler et al., 2018), they still lag behind the powerful self-attention in ViTs. Another line of works (Mehta & Rastegari, 2021; Chen et al., 2022b; Li et al., 2023; Cai et al., 2023; Shaker et al., 2023; Vasu et al., 2023a; Qin et al., 2024) proposes innovative attention mechanisms to address the limitation of standard attention (Vaswani, 2017) and blend convolutions to achieve a better balance between latency and performance. These attention mechanisms either reduce the number of queries and keys (Shaker et al., 2023; Qin et al., 2024), limit the attention span (Wan et al., 2023), or adopt linear attention (Cai et al., 2023), which may compromise performance to some extent.

In this work, we present the iFormer, a herd of lightweight models that integrates the strengths of both CNNs and ViTs, achieving a state-of-the-art balance between latency and accuracy. Specifically, we employ a hierarchical architecture consisting of four stages. In the earlier, high-resolution stages, we utilize fast convolution to extract local representations. To construct the convolutional block, we start with a "modern" ConvNeXt (Liu et al., 2022), which incorporates a series of design decisions inspired by ViTs. Then we progressively "lighten" the ConvNeXt to create a streamlined lightweight network, optimizing it for real-time mobile latency on an iPhone 13, in contrast to the FLOPs and parameters used in prior works (Mehta & Rastegari, 2021; Chen et al., 2022b). This results in a fast convolutional architecture with strong performance. To further enhance the dynamic properties and its ability to model long-range contexts, we incorporate self-attention in the later low-resolution stages. However, direct implementation of standard multi-head self-attention (MHA) brings notable memory overheads and slows down inference speed on mobile devices. We identify that the increased latency stems primarily from the reshaping operations in MHA. More analyses reveal that multiple attention heads behave similarly. Therefore, we propose a simple yet effective single-head modulation self-attention (SHMA), which significantly minimizes memory costs while preserving strong performance. Fig. 4 provides an illustration of SHMA. In detail, SHMA learns spatial context interactions through optimized self-attention. Concurrently, a parallel feature extraction branch is employed to capture informative features. Finally, we fuse the outputs of these two branches to facilitate a more flexible and dynamic exchange of information, compensating for the slight performance degradation of the single-head attention when compared to MHA.

Benefiting from the fast local representation capacity of convolution and the efficient global modeling proficiency of the proposed SHMA, iFormer outperforms existing pure lightweight CNNs and hybrid networks across multiple visual recognition tasks, including image classification, object detection, instance segmentation, and semantic segmentation. For instance, in the context of image classification as shown in Fig. 1, iFormer-M achieves a Top-1 accuracy of 80.4% with only 1.10 ms on an iPhone 13 without advanced training strategies such as knowledge distillation (Touvron et al., 2021a) or reparameterization (Ding et al., 2021). Notably, our model obtains a 0.5% improvement in Top-1 accuracy compared to the recent MNV4-Conv-M (Qin et al., 2024), while being 1.4× faster than FastViT-SA12 (Vasu et al., 2023a) with similar accuracy. These results demonstrate the effectiveness of the proposed network in capturing both local and global feature representations.

## 2 RELATED WORK

### 2.1 EFFICIENT CONVOLUTIONAL NETWORKS

In the past 2010s, computer vision was dominated by CNNs, and so were efficient networks. The first remarkable breakthrough in mobile CNNs is MobileNets (Howard et al., 2017), which hatches the concept of decomposing standard convolution into depthwise and pointwise counterparts. Subsequently, MobileNetV2 (Sandler et al., 2018) introduces an inverted residual bottleneck block to push the state-of-the-art for mobile models. Numerous studies have aimed to accelerate CNNs via various approaches, such as channel shuffle in ShuffleNet (Zhang et al., 2018; Ma et al., 2018) and cheap linear transformations in GhostNet (Han et al., 2020). Meanwhile, Neural architecture search

(NAS) has emerged as a method for automating the design of neural networks, optimizing for performance under resource constraints. EfficientNet (Tan & Le, 2019), MobileNetV3 (Howard et al., 2019), and FBNet (Wu et al., 2019) all achieve rather good performance. Besides, MobileOne (Vasu et al., 2023b) proposes to train a model using reparameterizable branches, which are merged during inference. Recently, following the revolution of ViTs, several methods reexamine the design spaces and training strategies (Liu et al., 2024) for mobile CNNs. For instance, RepViT (Wang et al., 2024) integrates efficient architectural designs from ViTs into MobileNetV3, outperforming existing lightweight CNNs. Other approaches, such as FocalNet (Yang et al., 2022a), Conv2Former (Hou et al., 2024), and EfficientMod (Ma et al., 2024), fuse features from context modeling and feature projection branches, also known as modulation mechanism, to enhance the model with dynamic properties analogous to attention. However, pure CNNs remain inherently spatially localized and their reliance on stationary weights restricts their flexibility. Although modulation can partially mitigate this limitation by enhancing dynamic capacity, they still exhibit deficiencies in building global interactions.

## 2.2 EFFICIENT VISION TRANSFORMERS

The success of Vision Transformer (Dosovitskiy et al., 2020) offers a compelling demonstration of the potential to apply transformer to computer vision tasks. Following this, ViT and its numerous variants (Liu et al., 2021a; Dong et al., 2022; Li et al., 2022a) sweep across various scenarios. However, the quadratic complexity of self-attention behind ViTs poses significant challenges for efficiency. The following researches seek to boost ViT efficiency through efficient attention mechanisms (Wang et al., 2021; Zhu et al., 2023; Hatamizadeh et al., 2023), model compression (Liu et al., 2021b; Zheng et al., 2022), knowledge distillation (Hao et al., 2021), and token reduction (Rao et al., 2021; Bolya et al., 2022). Recent studies further introduce ViTs into mobile applications. One mainstream of work combines efficient convolution and ViT to create lightweight hybrid networks (Mehta & Rastegari, 2022; Vasu et al., 2023a). MobileViT (Mehta & Rastegari, 2021) directly integrates MobileNetv2 blocks and ViT blocks, while Mobile-Former (Chen et al., 2022b) features a parallel design of MobileNet and ViT with a two-way bridge connecting the two. To further accelerate inference, some approaches replace the standard attention (Vaswani, 2017) with efficient variants within the hybrid networks. These include reducing the number of delegate tokens for computing attention (Pan et al., 2022), employing channel attention (Maaz et al., 2022), substituting projection in attention with efficient ghost modules (Ma et al., 2022), and utilizing linear attention mechanisms (Zhao et al., 2022). Besides manual designs, EfficientFormer (Li et al., 2022b; 2023) and MobileNetV4 (Qin et al., 2024) search for efficient architectures in a unified space encompassing both convolution operators and transformer operators. Another stream of work focuses on efficient attention mechanisms and directly employs them throughout the entire network (Shaker et al., 2023; Cai et al., 2023). For example, CMT (Guo et al., 2022) takes advantage of depth-wise convolution to downsample key and value to reduce computation. GhostNetV2 (Tang et al., 2022) applies two fully connected layers along the horizontal and vertical directions to compute attention, a decoupled version of MLP-Mixer (Tolstikhin et al., 2021). Recently, SHViT observes computational redundancy in the multi-head attention module and proposes to apply sing-head attention. In contrast to these existing approaches, we introduce a novel efficient attention module without sacrificing informative interactions, thereby maintaining strong representational capacity. Regarding attention design, ours is a bit similar to SHViT but is considerably superior as shown in Table 18 in the supplementary material. The key difference lies in the novel modulation attention. In addition, we explore efficient attention mechanisms in an on-device environment while SHViT focuses on general-purpose GPUs, fundamentally different hardware.

## 3 METHOD

We present the overall architecture of our iFormer in Fig. 4, which offers a Pareto-optimal accuracy-latency trade-off on mobile applications. Our exploration towards a streamlined lightweight network unfolds as follows: 1) establishing the baseline and measure metric in Sec. 3.1. 2) exploring acceleration techniques consisting of macro and micro designs in Sec. 3.2. 3) injecting global attention in Sec. 3.3. Finally, we create a new family of efficient hybrid vision transformers tailored for mobile applications in Sec. 3.3. A detailed trajectory illustrating the evolution from a general hierarchical CNN to a fast hybrid vision transformer is depicted in Fig. 2.

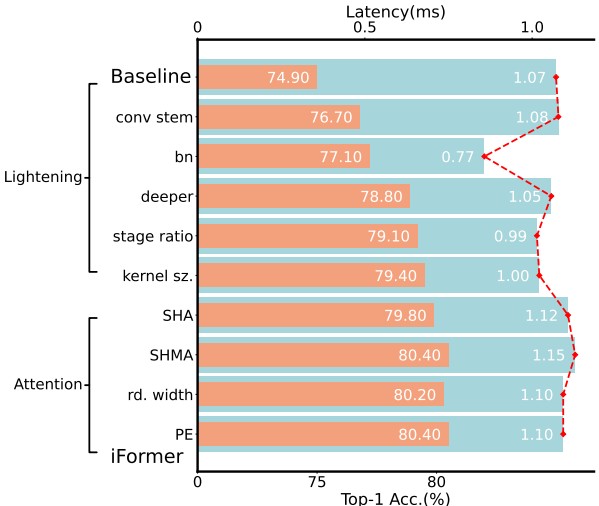

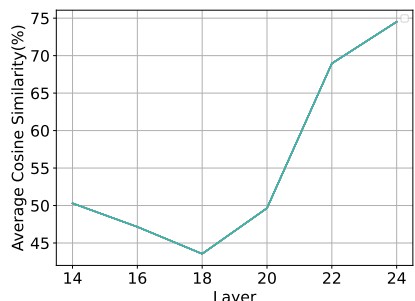

Figure 3: **The distribution of average cosine similarity among multiple heads within the MHA mechanism.** As the layer depth increases, the similarity goes higher.

Figure 2: **Illustration of the evolution from the ConvNeXt baseline towards the lightweight iFormer.** The orange bars are model accuracies and the light blue bars are model latencies. We also include a red latency outline for better visualization.

Table 1: **Latency comparison between multi-head and single-head baseline.**

| Models | Latency (ms) | Top-1 Acc. (%) |
|---|---|---|
| MHA Baseline | 1.40 | 79.9 |
| SHA Baseline | 1.12 (1.25×) | 79.8 |

## 3.1 PREPARING CONVNEXT

Our goal is to create an efficient multiscale network, where spatial dimensions of intermediate representations shrink as inference proceeds. In this hierarchical architecture, early network layers have larger spatial dimensions and fewer channels (e.g. 56×56×48), which renders them memory-bound. Highly optimized convolution is more appropriate for these layers. Guided by this principle, we choose a pure convolutional network as our base architecture, specifically ConvNeXt (Liu et al., 2022) which absorbed several key components from ViTs and competes favorably against ViTs. We gradually "lighten" the network to achieve a more favorable balance between latency and accuracy. For speed metric, we utilize on-device latency, measured on an actual iPhone 13 and compiled by Core ML Tools (CoreML), rather than FLOPs and parameter counts in previous methods (Mehta & Rastegari, 2021; Chen et al., 2022b; Zhang et al., 2022), which are not well correlated with latency. Regarding performance, we follow the training recipe in ConvNeXt while removing the layer scale to align prior methods (Li et al., 2022b; Wang et al., 2024) for a fair comparison. Please refer to Sec. B in the supplementary material for more details. To initiate our study, we systematically scale down the ConvNeXt by reducing the number of blocks and the width. This results in a lightweight model with a latency of 1.07 ms and a Top-1 accuracy of 74.9%, serving as our initial baseline.

## 3.2 LIGHTENING BASELINE

**Seeing Better with Early Convolutions** Following ViTs, ConvNeXt adopts an aggressive "patchify" strategy as the stem cell, specifically by splitting the input image into a series of non-overlapping patches via a 4x4 non-overlapping convolutional layer. However, some studies (Xiao et al., 2021; Chen et al., 2022a) indicate that an early convolutional stem can increase optimization stability and facilitate faster model convergence. Moreover, compared to general models, lightweight models typically have fewer parameters and a reduced capacity. An aggressive non-overlapping layer may lead to the premature loss of rich information. Consequently, we opt to replace the non-overlapping "patchify" stem with a stack of overlapping convolutional layers, as shown in Fig. 4. This modification elevates the top-1 accuracy to 76.7% with a neglectable increase in latency of 0.1 ms.

**Normalization** An obvious difference between ConvNeXt and previous CNNs is the normalization layer. ConvNeXt utilizes Layer Normalization (LN) (Ba et al., 2016), commonly used in Natural Language Processing (NLP), whereas the latter uses Batch Normalization (BN) (Ioffe, 2015). Albeit its superior performance, LN requires on-the-fly statistics calculation in inference along with

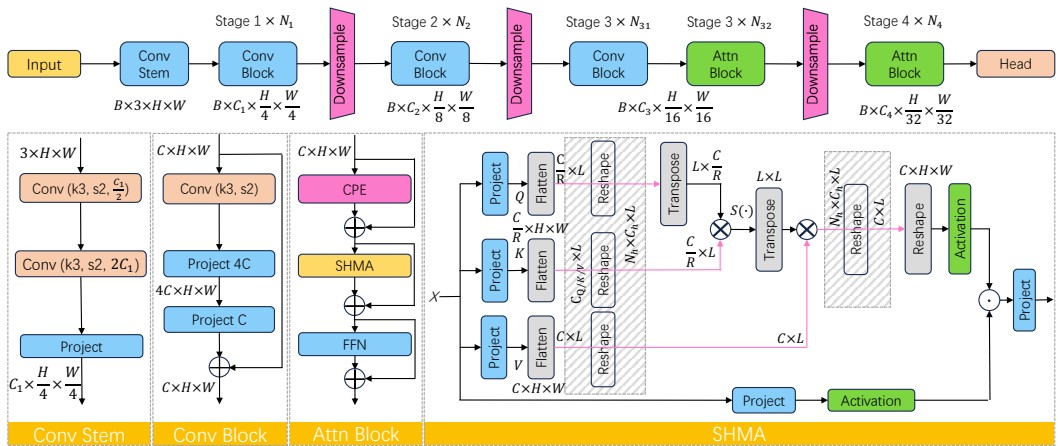

Figure 4: **Overview of iFormer architecture, detailed convolutional stem, block design, and SHMA.** The hatched area in SHMA indicates extra memory-intensive reshaping operations that are eliminated by SHMA. $S(\cdot)$ denotes the softmax function. $R$ is the ratio for reducing channels of query and key. It is set to 2 in iFormer. We omit BN following project or convolution for simplicity.

division and square root operations, leading to inefficiency on mobile hardware (Yang et al., 2022b). On the contrary, BN operates with fixed statistics during inference as an offline method and can be seamlessly fused with other linear operations, providing a "free lunch". This significantly reduces computational demands and memory overheads on mobile devices. Therefore, we substitute LN with BN throughout the network and merge it during inference. Additionally, we also substitute non-overlapping downsample layers with overlapping counterparts. These adjustments result in a reduction of overall latency to 0.77 ms while enhancing the Top-1 accuracy slightly to 77.10%.

**Going Deeper** There is considerable evidence indicating that increasing the depth of a model can enhance its capacity and yield performance benefits (Touvron et al., 2021b; Yang et al., 2022a). Most lightweight models typically stack more blocks to boost performance within constrained resources, as exemplified by the MobileNet series (Howard et al., 2019; Qin et al., 2024). In this study, we explore the potential of deepening ConvNeXt by increasing the number of blocks in each stage from (2,2,6,2) to (3,3,9,3). This increase in depth leads to a substantial improvement, raising the accuracy from 77.1% to 78.8%, although causing a temporary increase in latency to 1.05 ms.

**Stage Ratio** The stage ratio in ConNeXt is not optimized for lightweight models. A substantial number of depthwise convolutions in the early stages incurs significant memory transfer costs. Meanwhile, the presence of many blocks with a channel expansion ratio of 4 in the Feed-Forward Network (FFN) in the last stage, which already has a high channel dimension, imposes substantial computational demands. These factors lead to a sub-optimal allocation of computational resources. To address these issues, we propose reallocating more computational resources to the third stage while reducing memory access costs in the early stage. Specifically, the blocks in each stage are adjusted from (3,3,9,3) to (2,2,18,2). As expected, this achieves a better balance between latency and performance, with Top-1 accuracy increasing to 79.1% while enjoying a lower latency of 1.01ms.

**Kernel Size** Here we examine the effects of different kernel sizes in mobile settings and observe that utilizing a larger kernel size introduces nearly no latency burden, as shown in Table 2. So we maintain the convolutional kernel size at 7×7 in each basic block, consistent with ConvNeXt. Furthermore, previous approaches use a kernel size of 3×3 in the convolutional stem. This small receptive field may hinder feature representation during the early downsampling process. As previously noted, the early layers are memory-bound, allowing for opportunities to employ compute-intensive operations (*i.e.*, dense convolution). Therefore, we enlarge the kernel size of the dense

Table 2: **Latency under different convolutional kernel sizes.**

| Kernel Size | Latency (ms) |
|:-----------:|:------------:|
| 3×3 | 1.00 |
| 7×7 | 1.01 |

convolutional layer in the stem cell to 5×5. As illustrated in Fig. 2, this change has no impact on inference latency while enhancing Top-1 accuracy by 0.3%.

### 3.3 SINGLE-HEAD MODULATION ATTENTION

**Single-Head *vs*. Multi-Head**   ViTs typically apply MHA, which projects the queries, keys, and values multiple times with different learnable linear projections and performs multiple attention functions simultaneously. In practice, the multi-head mechanism requires the reshaping of feature maps first, causing large memory access and transfer costs. This can seriously impact inference latency, especially on resource-constrained mobile devices. To investigate this issue, we substitute the last half of the convolutional blocks in the third stage and all blocks in the last stage with standard ViT blocks, as depicted in Fig. 4. We refer to this hybrid network as the MHA baseline. Next, we build another network by substituting the MHA with Single-Head self-Attention (SHA), referring to it as the SHA baseline. The comparison is shown in Table 1. The SHA baseline shows a 1.25× acceleration over its MHA counterpart on the iPhone 13. This verifies that additional reshaping operations in MHA incur significant memory access costs, leading to a considerable decline in inference speed.

This naturally calls for optimizing MHA. Recent methods (Pan et al., 2022; Qin et al., 2024) primarily focus on downsampling the query or the key, which may hurt global attention capacity. Instead, we aim to reduce the redundant reshaping of MHA while preserving all token-to-token interactions. Previous works (Michel et al., 2019; Yun & Ro, 2024) indicate that a single attention head can approach the performance of multiple heads in general plain transformer models, such as DeiT. To investigate this on the mobile application, we analyze the average cosine similarity of multiple heads within the same layer of the aforementioned MHA baseline, which is a hierarchical lightweight network, and present our findings in Fig. 3. We clearly see that the average cosine similarity reaches 50% and even 75% in the final layer. Furthermore, the SHA baseline, as shown in Table 1, exhibits only a negligible accuracy drop of 0.1%. These suggest that SHA achieves a more favorable balance between accuracy and latency, obtaining an accuracy of 79.8% with a latency of 1.12 ms.

**Modulation Attention**   We further introduce a novel modulation attention to boost performance and strengthen flexibility in modeling, as illustrated in Fig. 4. Formally, we start from the abstracted modulation mechanism (Ma et al., 2024), similar to the gate mechanism Shazeer (2020). Assume we are given an input feature map $\mathbf{x} \in \mathbb{R}^{C \times H \times W}$ where $C$, $H$, and $W$ denote the channels, height, and width of the feature map. The modulated output can be written as follows:

$$\mathbf{x_o} = f(\mathbf{x}) \odot \text{ctx}(\mathbf{x}), \tag{1}$$

where $f(\cdot)$ denotes the feature mapping branch and $\text{ctx}(\cdot)$ is the context modeling branch. The output $\mathbf{x_o}$ is the fused features from both branches via efficient element-wise multiplication. The key idea of our approach is to modulate the feature using SHA instead of relying on convolutional layers, as seen in previous works (Yang et al., 2022a; Ma et al., 2024). Since SHA captures global interactions through self-attention, it excels in extracting rich contextual information and better controlling the flow of information. This process can be expressed as follows:

$$\text{ctx}(\mathbf{x}) = \text{SHA}(\mathbf{W^Q x}, \mathbf{W^K x}, \mathbf{W^V x}), \tag{2}$$

where $\mathbf{W^Q}, \mathbf{W^K}, \mathbf{W^V}$ are the project weights for query, key, and value, respectively. For simplicity, we omit the bias term. To minimize inference costs, we utilize a single projection layer in the feature mapping branch. To enhance expressivity and improve optimization stability, we apply individual nonlinear activation functions to both branches, as follows:

$$\mathbf{x_o} = \sigma(\mathbf{W^M x}) \odot \sigma(\text{ctx}(\mathbf{x})), \tag{3}$$

where $\sigma$ is the sigmoid function and $\mathbf{W^M}$ denotes the feature projection weight. We also experiment with various activation functions for modulation in Sec. 5 and observe that the sigmoid works rather well. Finally, the output from the modulation attention is projected in a manner as standard attention.

Equipped with Single-Head Modulation Attention (SHMA), our model improves the accuracy to 80.4% with an intermediate latency of 1.15 ms. This performance notably surpasses that of the recent MobileNetV4, which achieves an accuracy of 79.9%.

**Reducing Width**    Until now, we have developed a lightweight network that performs pretty well, but at a bit slow speed. To push the trade-off toward the state-of-the-art, we revise the width configuration in the SHMA. The modulation mechanism enriches the output by enabling more dynamic modeling in both spatial and channel dimensions, making it possible to use a weaker SHA and FFN. In light of this, we reduce the head dimension in the SHMA (*i.e.*, $\mathbf{W^Q}, \mathbf{W^K}$) to a small factor of the feature dimension, further details can be found in Table 15 in the supplementary material. Simultaneously, we shrink the expansion ratio in FFN following SHMA from 4 to 3. This process obtains a lower latency of 1.10 ms, although a slight drop of 0.2% in accuracy.

**Positional Embedding**    Last but not least, positional information plays a crucial role in self-attention as it regards input as a set of tokens. Adding positional embedding will help the attention learn permutation-variant features. We apply conditional positional encodings (CPE) (Chu et al., 2021) that are dynamically generated and conditioned on the local neighborhood of the input tokens, as illustrated in Fig. 4. The integration of CPE further enhances our model's performance, achieving a Top-1 accuracy of 80.4% with only 1.10 ms, establishing a state-of-the-art trade-off.

**iFormer**    The result of these modifications is an extremely fast and efficient hybrid network, which we denote *iFormer*. The overall architecture is depicted inFig. 4. It integrates fast local convolutional layers in the early stages that operate on higher resolution and global SHMA in later stages which processes lower resolution. Besides, we create a series of iFormer models tailored to various hardware resource constraints. For detailed architectural hyperparameters of these model variants, please refer to Table 15 in the supplementary material.

## 4 EXPERIMENTS

### 4.1 IMAGE CLASSIFICATION

**Settings.**    We first evaluate our models on classification on ImageNet-1K (Deng et al., 2009). To ensure a fair comparison with prior studies, we follow the previous training recipe (Touvron et al., 2021a; Liu et al., 2022) and train all models for 300 epochs with a standard image size of 224x224. Please refer to Sec. B in the supplementary material for details. Besides Top-1 validation accuracy, we also report the latency measured on an iPhone 13 with models compiled by Core ML Tools (CoreML) under a batch size of 1, as done in (Li et al., 2023; Wang et al., 2024; Vasu et al., 2023b). It's worth highlighting that we do not apply any advanced strategies such as distillation (Li et al., 2023) and reparameterization (Ding et al., 2021).

Table 3 summarizes a comparison between our iFormer and state-of-the-art lightweight models, organized by latency. iFormer demonstrates a Pareto-optimal trade-off between accuracy and latency. For example, iFormer-M obtains 80.4% top-1 accuracy with a latency of only 1.1 ms, surpassing recent MobileNetV4-Conv-M and RepViT-M1 by 0.5% and 1.0%, respectively. This is noteworthy considering that MobileNetV4 requires a longer training schedule (500 *vs.* 300) and takes a larger input resolution (256 *vs.* 224).

Table 4: **Results with distillation on ImageNet-1K.** * indicates the model is trained with a strong training strategy (*i.e.*, reparameterization).

| Model | Latency (ms) | Reso. | Epochs | Top-1 (%) |
|---|---|---|---|---|
| EfficientFormerV2-S1 (2023) | 1.02 | 224 | 300 | 79.0 |
| EfficientFormerV2-S1 (2023) | 1.02 | 224 | 450 | 79.7 |
| MobileViGv2-S*(2024) | 1.24 | 224 | 300 | 79.8 |
| FastViT-T12* (2023a) | 1.12 | 256 | 300 | 80.3 |
| RepViT-M1.1* (2024) | 1.04 | 224 | 300 | 80.7 |
| **iFormer-M** | **1.10** | 224 | 300 | **81.1** |
| SHViT-S4 (2024) | 1.48 | 224 | 300 | 80.2 |
| EfficientFormerV2-S2 (2023) | 1.60 | 224 | 300 | 81.6 |
| MobileViGv2-M(2024) | 1.70 | 224 | 300 | 81.7 |
| FastViT-SA12* (2023a) | 1.50 | 256 | 300 | 81.9 |
| EfficientFormerV2-S2 (2023) | 1.60 | 224 | 450 | 82.0 |
| RepViT-M1.5* (2024) | 1.54 | 224 | 300 | 82.3 |
| **iFormer-L** | **1.60** | 224 | 300 | **82.7** |

224). When compared to other recent models using reparameterization, including FastViT-T12, GhostNetV3-1.3×, and MobileOne-S3, iFormer-M achieves superior accuracy while maintaining lower latency. Moreover, iFormer outperforms various hybrid networks. Thanks to the efficient SHMA, iFormer-L achieves more outstanding performance than other attention variants, such as multi-query attention in MNV4-Hybrid-M, additive attention in SwiftFormer-L1, and linear attention in EfficientVIT-B1-r288.

Table 3: **Classification results on ImageNet-1K.** [†] indicates models that are trained with a variety of advanced training strategies including complex reparameterization, distillation, optimizer, and so on. We provide a more comprehensive comparison in Sec. G in the supplementary material.

| Model | Params (M) | GMACs | Latency ↓ (ms) | Reso. | Epochs | Top-1 (%) |
|---|---|---|---|---|---|---|
| MobileNetV2 1.0x (2018) | 3.4 | 0.30 | 0.73 | 224 | 500 | 72.0 |
| MobileNetV3-Large 0.75x (2019) | 4.0 | 0.16 | 0.67 | 224 | 600 | 73.3 |
| MNV4-Conv-S (2024) | 3.8 | 0.20 | 0.60 | 224 | 500 | 73.8 |
| **iFormer-T** | 2.9 | 0.53 | **0.60** | 224 | 300 | **74.1** |
| MobileNetV2 1.4x (2018) | 6.9 | 0.59 | 1.02 | 224 | 500 | 74.7 |
| MobileNetV3-Large 1.0x (2019) | 5.4 | 0.22 | 0.76 | 224 | 600 | 75.2 |
| SwiftFormer-XS (2023) | 3.5 | 0.60 | 0.95 | 224 | 300 | 75.7 |
| SBCFormer-XS (2024) | 5.6 | 0.70 | 0.79 | 224 | 300 | 75.8 |
| GhostNetV3 1.0x[†] (2024) | 6.1 | 0.17 | 0.99 | 224 | 600 | 77.1 |
| MobileOne-S2 (2023b) | 7.8 | 1.30 | 0.92 | 224 | 300 | 77.4 |
| RepViT-M1.0 (2024) | 6.8 | 1.10 | 0.85 | 224 | 300 | 78.6 |
| **iFormer-S** | 6.5 | 1.09 | **0.85** | 224 | 300 | **78.8** |
| EfficientMod-xxs (2024) | 4.7 | 0.60 | 1.29 | 224 | 300 | 76.0 |
| SBCFormer-S (2024) | 8.5 | 0.90 | 1.02 | 224 | 300 | 77.7 |
| MobileOne-S3 (2023b) | 10.1 | 1.90 | 1.16 | 224 | 300 | 78.1 |
| SwiftFormer-S (2023) | 6.1 | 1.00 | 1.12 | 224 | 300 | 78.5 |
| GhostNetV3 1.3x[†] (2024) | 8.9 | 0.27 | 1.24 | 224 | 600 | 79.1 |
| FastViT-T12 (2023a) | 6.8 | 1.40 | 1.12 | 256 | 300 | 79.1 |
| RepViT-M1.1 (2024) | 8.2 | 1.30 | 1.04 | 224 | 300 | 79.4 |
| MNV4-Conv-M (2024) | 9.2 | 1.00 | 1.08 | 256 | 500 | 79.9 |
| **iFormer-M** | 8.9 | 1.64 | **1.10** | 224 | 300 | **80.4** |
| Mobile-Former-294M (2022b) | 11.4 | 0.29 | 2.66 | 224 | 450 | 77.9 |
| MobileViT-S (2021) | 5.6 | 2.00 | 3.55 | 256 | 300 | 78.4 |
| MobileOne-S4 (2023b) | 14.8 | 2.98 | 1.74 | 224 | 300 | 79.4 |
| SBCFormer-B (2024) | 13.8 | 1.60 | 1.44 | 224 | 300 | 80.0 |
| GhostNetV3 1.6x[†] (2024) | 12.3 | 0.40 | 1.49 | 224 | 600 | 80.4 |
| EfficientViT-B1-r288 (2023) | 9.1 | 0.86 | 3.87 | 288 | 450 | 80.4 |
| FastViT-SA12 (2023a) | 10.9 | 1.90 | 1.50 | 256 | 300 | 80.6 |
| MNV4-Hybrid-M (2024) | 10.5 | 1.20 | 1.75 | 256 | 500 | 80.7 |
| SwiftFormer-L1 (2023) | 12.1 | 1.60 | 1.60 | 224 | 300 | 80.9 |
| EfficientMod-s (2024) | 12.9 | 1.40 | 2.57 | 224 | 300 | 81.0 |
| RepViT-M1.5 (2024) | 14.0 | 2.30 | 1.54 | 224 | 300 | 81.2 |
| **iFormer-L** | 14.7 | 2.63 | **1.60** | 224 | 300 | **81.9** |

**Results with distillation on ImageNet-1K.** We conducted rigorously fair training as the previous methods above. Recently, some works report enhanced performance leveraging more advanced training strategies. We investigate whether these training recipes can also improve iFormer. Following previous works (Li et al., 2023; Wang et al., 2024), we employ the RegNetY-16GF (Radosavovic et al., 2020) model with a top-1 accuracy of 82.9% as the teacher model for distillation. Our findings reveal that iFormer improves obviously over its counterpart without distillation. For example, iFormer-L shows a 1.0% increase under the same latency. iFormer also outperforms EfficientFormerV2-S2, despite the latter being trained with a $1.5\times$ longer schedule.

## 4.2 OBJECT DETECTION AND INSTANCE SEGMENTATION

To validate the effectiveness of iFormer on downstream tasks, we train Mask R-CNN (He et al., 2017) with iFormer as the backbone for 12 epochs ($1\times$), using the MMDetection toolkit (Chen et al., 2019). We also report backbone latency measured at a resolution of $512\times512$ on an iPhone 13. The results are presented in Table 5. In comparison to lightweight models, iFormer-M surpasses FastViT-SA12 by +1.9%/+2.0% in $AP^{box}$ /$AP^{mask}$ while running $1.32\times$ faster. iFormer-L also obtains +0.1%/+0.6% in $AP^{box}$ /$AP^{mask}$ than EfficientMod-S, which utilizes a convolutional modulation mechanism to learn dynamics similar to self-attention. Notably, EfficientMod-S operates $3.7\times$ slower when processing high-resolution input, underscoring that the proposed novel attention mechanism is more suitable for mobile networks. Meanwhile, when compared to general networks that are not optimized for mobile applications, iFormer demonstrates significant advantages. For instance, iFormer-L exceeds the performance of ConvNeXt-T with improvements of +1.2%/+1.4% in

Table 5: **Object detection & instance segmentation** results on MS COCO 2017 using Mask R-CNN. **Semantic segmentation** results on ADE20K using the Semantic FPN framework. We measure all backbone latencies with image crops of 512×512 on iPhone 13 by Core ML Tools. Failed indicated that the model runs too long to report latency by the Core ML.

| Backbone | Param (M) | Latency ↓ (ms) | Object Detection | | | Instance Segmentation | | | Semantic |
|---|---|---|---|---|---|---|---|---|---|
| | | | $AP^{box}$ | $AP^{box}_{50}$ | $AP^{box}_{75}$ | $AP^{mask}$ | $AP^{mask}_{50}$ | $AP^{mask}_{75}$ | mIoU |
| EfficientNet-B0 (2019) | 5.3 | 4.55 | 31.9 | 51.0 | 34.5 | 29.4 | 47.9 | 31.2 | - |
| ResNet18 (2016) | 11.7 | 2.85 | 34.0 | 54.0 | 36.7 | 31.2 | 51.0 | 32.7 | 32.9 |
| PoolFormer-S12 (2022) | 11.9 | 5.70 | 37.3 | 59.0 | 40.1 | 34.6 | 55.8 | 36.9 | 37.2 |
| EfficientFormer-L1 (2022b) | 12.3 | 3.50 | 37.9 | 60.3 | 41.0 | 35.4 | 57.3 | 37.3 | 38.9 |
| FastViT-SA12 (2023a) | 10.9 | 5.27 | 38.9 | 60.5 | 42.2 | 35.9 | 57.6 | 38.1 | 38.0 |
| RepViT-M1.1 (2024) | 8.2 | 3.18 | 39.8 | 61.9 | 43.5 | 37.2 | 58.8 | 40.1 | 40.6 |
| iFormer-M | 8.9 | 4.00 | 40.8 | 62.5 | 44.8 | 37.9 | 59.7 | 40.7 | 42.4 |
| ResNet50 (2016) | 25.5 | 7.20 | 38.0 | 58.6 | 41.4 | 34.4 | 55.1 | 36.7 | 36.7 |
| PoolFormer-S24 (2022) | 21.4 | 10.0 | 40.1 | 62.2 | 43.4 | 37.0 | 59.1 | 39.6 | 40.3 |
| ConvNeXt-T (Liu et al., 2022) | 29.0 | 13.6 | 41.0 | 62.1 | 45.3 | 37.7 | 59.3 | 40.4 | 41.4 |
| EfficientFormer-L3 (2022b) | 31.3 | 8.40 | 41.4 | 63.9 | 44.7 | 38.1 | 61.0 | 40.4 | 43.5 |
| RepViT-M1.5 (2024) | 14.0 | 5.00 | 41.6 | 63.2 | 45.3 | 38.6 | 60.5 | 41.5 | 43.6 |
| PVTv2-B1 (2022) | 14.0 | 27.00 | 41.8 | 64.3 | 45.9 | 38.8 | 61.2 | 41.6 | 42.5 |
| FastViT-SA24 (2023a) | 20.6 | 8.97 | 42.0 | 63.5 | 45.8 | 38.0 | 60.5 | 40.5 | 41.0 |
| EfficientMod-S (2024) | 32.6 | 24.30 | 42.1 | 63.6 | 45.9 | 38.5 | 60.8 | 41.2 | 43.5 |
| Swin-T (2021a) | 28.3 | Failed | 42.2 | 64.4 | 46.2 | 39.1 | 61.6 | 42.0 | 41.5 |
| iFormer-L | 14.7 | 6.60 | 42.2 | 64.2 | 46.0 | 39.1 | 61.4 | 41.9 | 44.5 |

$AP^{box}$ /$AP^{mask}$, while requiring fewer parameters and only 50% mobile latency, suggesting iFormer's efficient design in feature extraction and strong potential for mobile applications.

## 4.3 SEMANTIC SEGMENTATION

We conduct experiments on the ADE20K (Zhou et al., 2017) using the Semantic FPN (Kirillov et al., 2019), based on the MMSegmentation toolkit (Contributors, 2020). Thanks to its efficient attention design, iFormer outperforms all competing methods in mIoU with similar and much lower latency. For example, iFormer-L surpasses FastViT-SA24 by +3.5% in mIoU with a 1.36× faster inference speed. In addition, iFormer-M demonstrates superior mIoU compared to general networks, which typically exhibit substantially greater latency when processing higher-resolution inputs on mobile devices. Although PVTv2-B utilizes downsampled attention, it still requires 27 ms for latency. Similarly, Swin-T involves intensive operations in window partitioning, making it less suitable for mobile applications. Running at 6.6 ms, iFormer-L achieves +2.0% better mIoU than PVTv2-B1 and +3.0% better than Swin-T. These results suggest that the proposed attention mechanism offers significant benefits for tasks requiring the perception of fine-grained details.

## 5 ABLATION STUDIES

**Activation Function** Here we explore whether an activation function without an upper bound can enhance the SHMA by allowing neurons to express arbitrarily large values. We compare the widely used Sigmoid Linear Unit (SiLU) (Shazeer, 2020) with the sigmoid function and present the results in Table 6. Directly replacing the activation function in SHMA with SiLU will encounter diverging loss during training. The underlying cause is primarily attributed to the element-wise multiplication of the unbounded context branch. To address this, we replace Post-BN in SHMA with Pre-LN, as LN adaptively normalizes each token feature. The modified model experiences a slight decrease in accuracy but incurs an additional 0.07 ms latency, primarily brought by LN. The results suggest that the sigmoid function not only mitigates training instability but also facilitates better convergence.

Table 6: **Activation function comparison in SHMA.** Post-BN indicates that BN is applied after projection. Pre-LN means that LN is implemented before the projection, as in standard MHA (Vaswani, 2017).

| SHMA Setting | Params (M) | GMACs | Latency (ms) | Top-1 Acc. (%) |
|---|---|---|---|---|
| SiLU + Post-BN | 8.9 | 1.60 | 1.10ms | Diverged |
| SiLU + Pre-LN | 8.9 | 1.64 | 1.17ms | 80.3 |
| Sigmoid + Post-BN | 8.9 | 1.60 | 1.10ms | 80.4 |

**Choice of Conv v.s. ViT Blcoks**  In Section 3.3, we replace the convolutional blocks in Stages 3 and 4 with the proposed SHMA block. We provide further ablation studies on the choice of ratio for the ViT blocks. Specifically, We choose the model after enlarging the kernel size as a starting point, then we progressively replace the convolutional blocks in Stages 3 and 4. We do not modify Stages 1 and 2 as their larger spatial dimensions would considerably increase the memory requirements for the self-attention mechanism.

Table 7: **Different ratio of ViT Block.**

| Ratio Setting | Params (M) | GMACs | Latency (ms) | Top-1 Acc. (%) |
|---|---|---|---|---|
| Baseline | 9.4M | 1760M | 1.0ms | 79.4 |
| Replacing 22% Conv Blocks in Stage 3 as SHA | 9.1M | 1724M | 1.02ms | 79.5 |
| Replacing 22% Conv Blocks in Stage 3 as SHMA | 9.2M | 1739M | 1.04ms | 79.6 |
| Replacing 50% Conv Blocks in Stage 3 as SHA | 8.8M | 1689M | 1.04ms | 79.5 |
| Replacing 50% Conv Blocks in Stage 3 as SHMA | 8.9M | 1712M | 1.07ms | 79.8 |
| Replacing 78% Conv Blocks in Stage 3 as SHA | 8.3M | 1635M | 1.12ms | 79.3 |
| Replacing 78% Conv Blocks in Stage 3 as SHMA | 8.5M | 1685M | 1.17ms | 79.6 |
| Replacing 100% Conv Blocks in Stage 3 as SHA | 7.9M | 1599M | 1.17ms | 78.1 |
| Replacing 100% Conv Blocks in Stage 3 as SHMA | 8.3M | 1665M | 1.25ms | 79.0 |
| Replacing 100% Conv Blocks in Stage 3 as SHMA and 100% in Stage 4 | 10.0M | 1792M | 1.15ms | 80.4 |

We present our findings in Table 7. Given that Stage 4 contains only two blocks, we do not conduct further splitting for the ratio. As illustrated in Table 7, although the ViT block has lower FLOPs, it still incurs increased runtime. Substituting all the convolutional blocks in Stage 3 results in the worst performance and the highest latency. Instead, by replacing half of the convolutional blocks in the third stage and all blocks in the final stage, we can better integrate these two operators, thus achieving a favorable trade-off between accuracy and latency.

**Scaling to Larger Model**  Although iFormer is designed for mobile-device applications, the combination of fast local representation capacity of convolution and the efficient global modeling proficiency of the proposed SHMA enables its scalability for a broader range of applications. To demonstrate the scalability of iFormer, we developed a larger model named iFormer-H with 99M parameters and trained it for 300 epochs following the same strategy outlined in Section B. It is important to note that we add drop path and layer scale, which are commonly used in the training of larger models (Liu et al., 2022; Tu et al., 2022; Shi, 2024).

We summarize the results in Table 8. A highlight from the results is that iFormer is not specifically designed or trained for this scale. Despite this, iFormer-H outperforms ConvNeXt, achieving a 1.0% increase in accuracy while maintaining a similar number of FLOPs.

Table 8: **Scaling to the larger model with 99M parameters.**

| Model | Params (M) | GMACs (G) | Top-1 Acc. (%) |
|---|---|---|---|
| ConvNeXt-Base (2022) | 89 | 15.4 | 83.8 |
| TransNeXt-Base (2024) | 90 | 18.4 | 84.8 |
| iFormer-H | 99 | 15.5 | 84.8 |
| MaxViT-Base (2022) | 120 | 24.0 | 84.9 |

Additionally, it demonstrates comparable performance to TransNeXt-Base, despite utilizing fewer FLOPs. These findings indicate the potential for broader applications of iFormer. We plan to explore larger models suitable for mobile devices in future work. Further ablation studies can be found in Sec. C in the supplementary material.

## 6  CONCLUSION

This work proposes iFormer, which integrates highly optimized convolutional operations for the early layers alongside a novel and efficient single-head modulation attention for the later layers. iFormer achieves SOTA Pareto-front in terms of Top-1 accuracy and mobile latency. We also validate the effectiveness of iFormer on downstream dense prediction tasks, including COCO object detection, instance segmentation, and ADE20K semantic segmentation. These inspiring results highlight the potential for mobile applications. We hope iFormer can facilitate the application of artificial intelligence on more mobile devices. In future work, we will seek to alleviate inference bottlenecks sociated with high-resolution images. Meanwhile, we plan to optimize iFormer for more hardware platforms, such as Android devices and NVIDIA Jetson Nano.

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
