## A    APPENDIX

## B    EXPERIMENTAL SETTINGS

### B.1    IMAGE CLASSIFICATION

Table 9: **ImageNet-1K training settings**.

| training config | iFormer-T/S/M/L/H |
|---|---|
| resolution | $224^2$ |
| weight init | trunc. normal (0.2) |
| optimizer | AdamW |
| base learning rate | 4e-3 (T/S/M/L) 8e-3 |
| weight decay | 0.05 |
| optimizer momentum | $\beta_1, \beta_2{=}0.9, 0.999$ |
| batch size | 4096 [T/S/M/L] 8192 [H] |
| training epochs | 300 |
| learning rate schedule | cosine decay |
| warmup epochs | 20 |
| warmup schedule | linear |
| layer-wise lr decay | None |
| randaugment | (9, 0.5) |
| mixup | 0.8 |
| cutmix | 1.0 |
| random erasing | 0.25 |
| label smoothing | 0.1 |
| stochastic depth | 0.0 [T/S/M] 0.1 [L] 0.6 [H] |
| layer scale | None [T/S/M/L] 1e-6 [H] |
| head init scale | None |
| gradient clip | None |
| exp. mov. avg. (EMA) | None |

We mainly follow the training recipe of ConvNeXt, while removing stochastic depth, layer scale, and exponential moving average to ensure a fair comparison with prior works. The models are trained for 300 epochs on 8 NVIDIA GPUs with a total batch size of 4096. We employ the same learning rate across all models. It is possible to further improve performance by adjusting the learning rates for different model variants, which we will explore in the future.

For distillation, we use the RegNetY-16GF model as the teacher model and apply a hard distillation loss, following the approach of DeiT (Touvron et al., 2021a). During inference, the average output of the classification head and the distillation head is used as the final output.

### B.2    OBJECT DETECTION AND SEMANTIC SEGMENTATION

For object detection experiments, we train MaskR-CNN models on the COCO 2017 dataset for 12 epochs using standard training settings from the MMDetection toolkit.

For semantic segmentation experiments, we train Semantic FPN models on the ADE20K dataset for 40,000 iterations using standard training settings from the MMSegmentation toolkit. The input images are cropped to a resolution of 512×512 during training.

For backbone latency, we keep the same input size as training (*i.e.*, 512×512) and measure the mobile latency on an iPhone 13 compiled by Core ML Tools.

## C    MORE ABLATION STUDIES

**Different Ways for Reducing Latency**    Here we provide a comparison of different methods for reducing latency, contrasting them with the approach discussed in Sec. 3.3. Specifically, we reduce the baseline latency to similar latency by directly removing blocks, cutting down FFN expansion width,

and reducing both attention head dimension and FFN expansion dimension simultaneously. From the results in Table 10, we observe that the removal of a single block in the final stage can lead to a severe drop in accuracy (-0.7%), indicating that greater depth enhances the model's capacity. Concurrently reducing all FFN expansion widths causes a non-trivial performance degradation (-0.6%).

In contrast, we observe that an orchestrated reduction in both attention head and FFN expansion dimensions yields a milder accuracy decline (-0.2%). These results demonstrate that a comprehensive reduction across different components offers better flexibility and performance.

Table 10: **Different ways for reducing latency.**

| Reducing Setting | Params (M) | GMACs | Latency (ms) | Top-1 Acc. (%) |
|---|---|---|---|---|
| Baseline | 10.0 | 1.79 | 1.15 | 80.4 |
| Number of Blocks | 8.4 | 1.70 | 1.07 | 79.7 |
| FFN Width | 8.6 | 1.62 | 1.07 | 79.8 |
| Attn. Head and FFN Width | 8.9 | 1.64 | 1.10 | 80.2 |

**Depthwise Convlution in FFN**   Recent works (Cai et al., 2023; Qin et al., 2024) attempt to insert a depthwise convolution (DW Conv) within the FFN to perform spatial mixing on the expanded features activations. We hypothesize that implementing more effective spatial mixing before the FFN diminishes its significance. In our iFormer, depthwise convolution with a kernel size of 7 is employed for spatial modeling in the early layers, while a powerful SHMA is utilized in the later layers. This approach provides a significantly enhanced spatial mixing capacity than previous methods.

As shown in Table 11, enhancing all FFN with depthwise convolution, including those within the convolutional blocks, results in a +14% increase in FLOPs and an additional latency cost of 0.33 ms. This increase is expected since the intermediate layers in the FFN possess an

Table 11: **Comparison of FFN with and without depthwise convolution.**

| DW Conv in FFN | Params (M) | GMACs | Latency (ms) | Top-1 Acc. (%) |
|---|---|---|---|---|
| with | 9.6 | 1.83 | 1.43 | 80.5 |
| w/o. | 8.9 | 1.60 | 1.10 | 80.4 |

expanded feature dimension. However, the Top-1 accuracy only exhibits a marginal improvement of +0.1%.

**Training for Longer Schedule**   Another commonly used advanced training is an extended schedule (450 *vs.* 300). Here we provide additional experiments for both image classification and downstream tasks where we train iFormer with distillation for 450 epochs. To ensure a fair comparison with previous methods, we develop a larger model dubbed as iFormer-L2. We report the image

Table 12: **Training with distillation for 450 epochs on ImageNet-1K.**

| Model | Params (M) | Latency (ms) | Reso. | Epochs | Top-1 (%) |
|---|---|---|---|---|---|
| ConvNeXt-B (2022) | 89.0 | 7.54 | 224 | 300 | 83.8 |
| EfficientFormerV2-L (2023) | 26.1 | 2.40 | 224 | 450 | 83.5 |
| **iFormer-L2** | **24.5** | **2.30** | 224 | 450 | **83.9** |

classification results on the ImageNet-1k dataset in Table 12. It shows that training iFormer-L2 for 450 epochs yields improved performance, obtaining a Top-1 accuracy of 83.9%, even surpassing the ConvNeXt-Base model.

Table 13: **Object detection & Semantic segmentation results using backbone pretrained for 450 epochs.**

| Backbone | Param (M) | Latency ↓ (ms) | Pretrain Epochs | Object Detection | | | Instance Segmentation | | | Semantic |
|---|---|---|---|---|---|---|---|---|---|---|
| | | | | $AP^{box}$ | $AP^{box}_{50}$ | $AP^{box}_{75}$ | $AP^{mask}$ | $AP^{mask}_{50}$ | $AP^{mask}_{75}$ | mIoU |
| ResNet50 (2016) | 25.5 | 7.20 | 300 | 38.0 | 58.6 | 41.4 | 34.4 | 55.1 | 36.7 | 36.7 |
| PoolFormer-S24 (2022) | 21.4 | 12.30 | 300 | 40.1 | 62.2 | 43.4 | 37.0 | 59.1 | 39.6 | 40.3 |
| ConvNeXt-T (Liu et al., 2022) | 29.0 | 12.6 | 300 | 41.0 | 62.1 | 45.3 | 37.7 | 59.3 | 40.4 | 41.4 |
| EfficientFormer-L3 (2022b) | 31.3 | 8.40 | 300 | 41.4 | 63.9 | 44.7 | 38.1 | 61.0 | 40.4 | 43.5 |
| RepViT-M1.5 (2024) | 14.0 | 5.00 | 300 | 41.6 | 63.2 | 45.3 | 38.6 | 60.5 | 41.5 | 43.6 |
| PVTv2-B1 (2022) | 14.0 | 27.00 | 300 | 41.8 | 64.3 | 45.9 | 38.8 | 61.2 | 41.6 | 42.5 |
| FastViT-SA24 (2023a) | 20.6 | 8.97 | 300 | 42.0 | 63.5 | 45.8 | 38.0 | 60.5 | 40.5 | 41.0 |
| EfficientMod-S (2024) | 32.6 | 24.30 | 300 | 42.1 | 63.6 | 45.9 | 38.5 | 60.8 | 41.2 | 43.5 |
| Swin-T (2021a) | 28.3 | Failed | 300 | 42.2 | 64.4 | 46.2 | 39.1 | 61.6 | 42.0 | 41.5 |
| iFormer-L | 14.7 | 6.60 | 300 | 42.2 | 64.2 | 46.0 | 39.1 | 61.4 | 41.9 | 44.5 |
| EfficientFormerV2-L (2023) | 26.1 | 12.5 | 450 | 44.7 | 66.3 | 48.8 | 40.4 | 63.5 | 43.2 | 45.2 |
| iFormer-L2 | 24.5 | 9.06 | 450 | 44.6 | 66.7 | 49.1 | 41.1 | 64.0 | 44.1 | 46.2 |

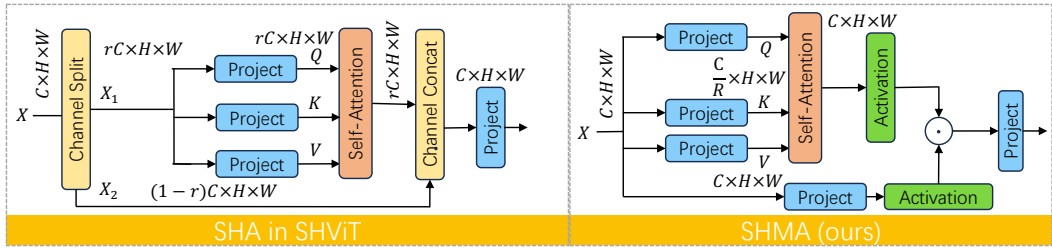

Figure 5: **Comparison of SHMA and SHA in SHViT.** In SHViT, $rC$ channels are utilized for spatial attention, where $r$ is set to $\frac{1}{4.67}$. SHMA projects the input into a higher dimension of $\frac{1}{2}$C (i.e., R=2) and avoids split and concatenation operations.

Furthermore, we integrate iFormer-L2 into the Mask-RCNN and Semantic FPN framework for downstream tasks. As anticipated, the model with the more powerful iFormer-L2 backbone achieves SOTA performance, obtaining a significant enhancement over models pretrained for 300 epochs. It also outperforms its EfficientFormerV2-L counterpart by +0.7% in AP$^{\text{mask}}$ and +1.0% in mIoU, while being 1.4× faster. These experiments collectively show that advanced training strategies can be easily employed to improve the performance of iFormers.

## D  RELATION TO SHVIT

We clarify the difference between SHA in iFormer and its counterpart in SHViT (Yun & Ro, 2024) from the following two aspects: First, in terms of motivation, iFormer explores efficient attention mechanisms specifically tailored for the on-device environment, whereas SHViT is geared towards general-purpose GPUs, which may exhibit different hardware characteristics. Second, in terms of methodology, as shown in Fig. 5, we utilize single-head attention with more channels ($R$ is set to 2.), while SHViT employs fewer than 1/4 of channels for attention. The reduced number of channels can result in a lower rank of the attention matrix, potentially degrading its expressiveness. Additionally, the split and concatenate operations in SHViT introduce extra runtime.

Table 14: **Process of converting SHA in iFormer towards SHViT.** Intermediate models are only measured by latency.

| Modification | Params(M) | GMACs | Latency (ms) | Top-1(%) |
|---|---|---|---|---|
| SHA Baseline without Modulation | 9.9M | 1758M | 1.12ms | 79.4 |
| + split | 9.9M | 1758M | 1.18ms | - |
| + attention on 1/4 channels | 8.3M | 1547M | 1.02ms | - |
| + concat | 8.7M | 1579M | 1.11ms | 79.5 |

We also conduct a more fair comparison with SHViT. We start from the SHA baseline referenced in Table 1, specifically denoted as 'SHA' in Figure 2. The transition to SHViT involves the following steps: 1) splitting the input into two smaller tensors, $X_1$ and $X_2$, along the channel dimension; 2) applying single-head attention to the tensor $X_1$, which contains fewer than 1/4 of channels present in the original input tensor; and 3) concatenating the attention output with the residual input $X_2$. As summarized in Table 14, split and concatenate operations introduce additional runtime. Furthermore, the performance of the SHA in the SHViT exhibits a decline compared to its counterpart in iFormer under similar latency conditions (79.8 v.s. 79.5). This degraded performance may be attributed to the reduced number of channels in the attention mechanism.

## E  ARCHITECTURE DETAILS

In Table 15, we show the different architecture configurations of the iFormer model variants.

## F  IFORMER FOR HIGHER RESOLUTION

Self-attention exhibits quadratic complexity with respect to the number of tokens, *i.e.*, the resolution of the input image. This issue is exacerbated in dense prediction tasks, which usually require high-

Table 15: **iFormer architecture configurations.** BN stands for Batch Normalization. SHMA stands for Singe-Head Modulation Attention. DW stands for Depthwise convolution. s and d means the stride and output dimension in convolution. hd denotes the head dimension in SHMA and the number of attention heads in all variants is 1. r means the expansion ratio in FFN.

| | Output Size (Downs. Rate) | iFormer-T | iFormer-S | iFormer-M | iFormer-L |
|---|---|---|---|---|---|
| Stem | 56×56 (4×) | [Conv-BN-GELU 5×5 s2 d16] × 1 | [Conv-BN-GELU 5×5 s2 d16] × 1 | [Conv-BN-GELU 5×5 s2 d24] × 1 | [Conv-BN-GELU 5×5 s2 d24] × 1 |
| | | [Conv-BN-GELU 5×5 s2 d64 / Conv-BN 1×1 s1 d32] × 1 | [Conv-BN-GELU 5×5 s2 d64 / Conv-BN 1×1 s1 d32] × 1 | [Conv-BN-GELU 5×5 s2 d96 / Conv-BN 1×1 s1 d48] × 1 | [Conv-BN-GELU 5×5 s2 d96 / Conv-BN 1×1 s1 d48] × 1 |
| Stage 1 | 56×56 (4×) | [Conv-BN 7×7 s1 d32 / Conv-BN-GELU 1×1 s1 d96 / Conv-BN 1x1 s1 d32] × 2 | [Conv-BN 7×7 s1 d32 / Conv-BN-GELU 1×1 s1 d128 / Conv-BN 1x1 s1 d32] × 2 | [Conv-BN 7×7 s1d48 / Conv-BN-GELU 1×1 s1 d192 / Conv-BN 1x1 s1 d48] × 2 | [Conv-BN 7×7 s1 d48 / Conv-BN-GELU 1×1 s1 d192 / Conv-BN 1x1 s1 d48] × 2 |
| Stage 2 | 28×28 (8×) | [Conv-BN 3×3 s2 d64] × 1 | [Conv-BN 3×3 s2 d64] × 1 | [Conv-BN 3×3 s2 d96] × 1 | [Conv-BN 3×3 s2 d96] × 1 |
| | | [Conv-BN 7×7 s1 d64 / Conv-BN-GELU 1×1 s1 d192 / Conv-BN 1x1 s1 d64] × 2 | [Conv-BN 7×7 s1 d64 / Conv-BN-GELU 1×1 s1 d256 / Conv-BN 1x1 s1 d64] × 2 | [Conv-BN 7×7 s1 d96 / Conv-BN-GELU 1×1 s1 d384 / Conv-BN 1x1 s1 d96] × 2 | [Conv-BN 7×7 s1 d96 / Conv-BN-GELU 1×1 s1 d384 / Conv-BN 1x1 s1 d96] × 2 |
| Stage 3 | 14×14 (16×) | [Conv-BN 3×3 s2 d128] × 1 | [Conv-BN 3×3 s2 d176] × 1 | [Conv-BN 3×3 s2 d192] × 1 | [Conv-BN 3×3 s2 d256] × 1 |
| | | [Conv-BN 7×7 s1 d128 / Conv-BN-GELU 1×1 s1 d384 / Conv-BN 1×1 s1 d128] × 6 | [Conv-BN 7×7 s1 d176 / Conv-BN-GELU 1×1 s1 d704 / Conv-BN 1x1 s1 d176] × 9 | [Conv-BN 7×7 s1 d192 / Conv-BN-GELU 1×1 s1 d768 / Conv-BN 1x1 s1 d192] × 9 | [Conv-BN 7×7 s1 d256 / Conv-BN-GELU 1×1 s1 d1024 / Conv-BN 1x1 s1 d256] × 8 |
| | | [CPE 3×3 / SHMA hd64 / FFN r2] × 3 | [CPE 3×3 / SHMA hd88 / FFN r3] × 3 | [CPE 3×3 / SHMA hd96 / FFN r3] × 4 | [CPE 3×3 / SHMA hd128 / FFN r3] × 8 |
| | | [Conv-BN 7×7 s1 d128 / Conv-BN-GELU 1×1 s1 d384 / Conv-BN 1x1 s1 d128] × 1 | [Conv-BN 7×7 s1 d176 / Conv-BN-GELU 1×1 s1 d704 / Conv-BN 1×1 s1 d176] × 1 | [Conv-BN 7×7 s1 d192 / Conv-BN-GELU 1×1 s1 d768 / Conv-BN 1×1 s1 d192] × 1 | [Conv-BN 7×7 s1 d256 / Conv-BN-GELU 1×1 s1 d1024 / Conv-BN 1×1 s1 d256] × 1 |
| Stage 4 | 7×7 (32×) | [Conv-BN 3×3 s2 d256] × 1 | [Conv-BN 3×3 s2 d320] × 1 | [Conv-BN 3×3 s2 d384] × 1 | [Conv-BN 3×3 s2 d384] × 1 |
| | | [CPE 3×3 / SHMA hd64 / FFN r2] × 2 | [CPE 3×3 / SHMA hd80 / FFN r3] × 2 | [CPE 3×3 / SHMA hd96 / FFN r3] × 2 | [CPE 3×3 / SHMA hd96 / FFN r3] × 2 |
| Params (M) | | 2.9 | 6.5 | 8.9 | 14.7 |
| GMacs | | 0.53 | 1.09 | 1.64 | 2.63 |

Table 16: **Comparison of different attention designs in iFormer-M.** For the sake of simplicity, we exclude other blocks that are not related to attention. ws is the window size for window attention.

| | Attention | SHMA | Hybrid SHMA | Chunk Hybrid SHMA |
|---|---|---|---|---|
| Stage 3 | 14×14 (16×) | | [CPE 3×3 / Window Partitioning, ws16 / Window SHMA hd96, ws16 / FFN r3] × 1 | [CPE 3×3 / Chunk Window Partitioning, ws16 / Window SHMA hd96, ws16 / FFN r3] × 1 |
| | | [CPE 3×3 / SHMA hd96 / FFN r3] × 4 | [CPE 3×3 / Window SHMA hd96, ws16 / FFN r3] × 2 | [CPE 3×3 / Window SHMA hd96, ws16 / FFN r3] × 2 |
| | | | [CPE 3×3 / Window Reversing, ws16 / SHMA hd96 / FFN r3] × 1 | [CPE 3×3 / Chunk Window Reversing, ws16 / SHMA hd96 / FFN r3] × 1 |
| Stage 4 | 7×7 (32×) | | [CPE 3×3 / Window Partitioning, ws16 / Window SHMA hd96 / FFN r3] × 1 | [CPE 3×3 / Chunk Window Partitioning, ws16 / Window SHMA hd96 / FFN r3] × 1 |
| | | [CPE 3×3 / SHMA hd64 / FFN r2] × 2 | [CPE 3×3 / Window Reversing, ws16 / SHMA hd64 / FFN r3] × 1 | [CPE 3×3 / Chunk Window Reversing, ws16 / SHMA hd64 / FFN r3] × 1 |

resolution input such as 512×512 in semantic segmentation and generate a large amount of 1024 image tokens even in the third stage. Consequently, this will cause huge memory and computation costs in mobile devices.

To mitigate these issues, we resort to window attention as proposed in Swin (Liu et al., 2021a). However, default window attention only performs local self-attention within windows, thus lacking interactions between tokens from different windows which will impair modeling capacity. Swin introduces shifted window attention to alleviate this limitation. Unfortunately, the shifting operation inevitably incurs additional memory costs. In contrast to Swin, we implement

Table 17: **Latency comparison of different attention mechanisms.**

| Attention | Resolution | Latency (ms) |
|---|---|---|
| SHMA | 224 | 1.10 |
| SHMA | 512 | Failed |
| Hybrid SHMA | 512 | 11.46 |
| CC Hybrid SHMA | 512 | 4.0 |

a hybrid attention design. Specifically, we compute window attention within windows, except for the last attention block in each stage. This approach enables iFormer to capture more global features essential for dense prediction tasks. At the same time, since window partitioning and reversing also incur memory access costs, we minimize the usage of them to once per stage. We replace the standard SHMA in iFormer with a hybrid window SHMA, as shown in Table 16.

From the latency comparison in Table 17, we see that simply applying SHMA will encounter a memory bottleneck on mobile devices. Instead, our hybrid SHMA can significantly reduce memory access costs, achieving a mobile latency of 11.46 ms.

However, hybrid SHMA still lags much behind the recent FastViT-SA12, which has a latency of 5.27 ms. We identify the speed bottleneck as stemming from the window partitioning and reversing operations, even though we only implement them once in each stage. As the feature map size increases, the reshaping involved in these operations demands considerable memory, thereby slowing inference in resource-constrained mobile devices.

To address this issue, we propose a method called "Channel Chunking" (CC). Formally, given a 2D input feature map $\mathbf{x} \in \mathbb{R}^{C \times H \times W}$, the standard window partitioning divides the feature map into $\frac{H}{P} \times \frac{W}{P}$ non-overlapped regions, each corresponding to a window that contains $P \times P$ feature vectors. This step is accomplished by reshaping x as $\mathbf{x^P} \in \mathbb{R}^{\frac{HW}{P^2} \times C \times P \times P}$. Then we apply SHMA within each window.

To reduce the memory requirements associated with reshaping, we propose to split the feature map $\mathbf{x}$ along the channel dimension into a series of smaller chunks as follows:

$$\mathbf{x_1^S}, ..., \mathbf{x_n^S} = \text{Chunking}(\mathbf{x}), \tag{4}$$

where K is the chunk size, $n = \frac{C}{K}$ is the number of chunks. We set n=16 for the input image of $512 \times 512$ in our object detection and semantic segmentation experiments. Then we apply window partitioning sequentially to these smaller chunks and concatenate them. This process can be mathematically expressed as follows:

$$\mathbf{x^P} = \text{Concat}(\mathbf{x_i^P}, ..., \mathbf{x_n^P}),$$
$$\text{where} \quad \mathbf{x_i^P} = \text{WindowPartitioning}(\mathbf{x_i^S}), \tag{5}$$

These smaller chunks can be processed rapidly. As shown in Table 17, the chunking strategy allows the model to achieve $2.9\times$ speed up in inference speed. Correspondingly, the window reversing operation is performed by reshaping multiple windows $\mathbf{x^P} \in \mathbb{R}^{\frac{HW}{P^2} \times C \times P \times P}$ into a 2D feature map $\mathbf{x} \in \mathbb{R}^{C \times H \times W}$. These results demonstrate that our proposed Channel Chunking Hybrid SHMA significantly enhances the iFormer's ability to process high-resolution images efficiently.

**Computation Complexity** Given an input $\mathbf{x} \in \mathbb{R}^{C \times H \times W}$ and a window size of P $\times$ P, as detailed in Section E, the computational complexity of iFormer is as follows:

$$\Omega(\text{SHMA}) = 4HWC^2(\text{QKV and output projection}) +$$
$$HWC(\text{element-wise product of modulation}) + \tag{6}$$
$$2P^2HWC(\text{self-attention}),$$

$$\Omega(\text{FFN}) = 8HWC^2. \tag{7}$$

In image classification, we do not utilize window attention since the feature size is $14 \times 14$ in stage 3 (it equals to the window attention when P=14). In downstream tasks, we adopt a window size of P=16.

## G    COMPREHENSIVE COMPARISON

In Table 18, we provide a more comprehensive comparison between iFormer and other lightweight models on ImageNet-1k classification.

Table 18: **Comprehensive comparison between iFormer and the previously proposed models on ImageNet-1K.** Failed indicated that the model runs too long to report latency by the Core ML, often caused by excessive memory access.

| Model | Params (M) | GMACs | Latency ↓ (ms) | Reso. | Epochs | Top-1 (%) |
|---|---|---|---|---|---|---|
| MobileNetV2 1.0x (2018) | 3.4 | 0.30 | 0.73 | 224 | 500 | 72.0 |
| SHViT-S1 (2024) | 6.3 | 0.24 | 0.74 | 224 | 300 | 72.8 |
| MobileNetV3-Large 0.75x (2019) | 4.0 | 0.16 | 0.67 | 224 | 600 | 73.3 |
| MNV4-Conv-S (2024) | 3.8 | 0.20 | 0.60 | 224 | 500 | 73.8 |
| **iFormer-T** | 2.9 | 0.53 | **0.60** | 224 | 300 | **74.1** |
| ShuffleNetV2 1.0× (2018) | 2.3 | 0.15 | 0.74 | 224 | 300 | 69.4 |
| MobileNetV2 1.4x (2018) | 6.9 | 0.59 | 1.02 | 224 | 500 | 74.7 |
| MobileNetV3-Large 1.0x (2019) | 5.4 | 0.22 | 0.76 | 224 | 600 | 75.2 |
| SwiftFormer-XS (2023) | 3.5 | 0.60 | 0.95 | 224 | 300 | 75.7 |
| SBCFormer-XS (2024) | 5.6 | 0.70 | 0.79 | 224 | 300 | 75.8 |
| GhostNetV3 1.0x$^{\dagger}$ (2024) | 6.1 | 0.17 | 0.99 | 224 | 600 | 77.1 |
| EfficientNet-B0 (2019) | 5.3 | 0.39 | 0.89 | 224 | 350 | 77.1 |
| MobileOne-S2 (2023b) | 7.8 | 1.30 | 0.92 | 224 | 300 | 77.4 |
| LowFormer-B0 (2024) | 14.1 | 0.94 | 1.45 | 224 | 300 | 78.4 |
| CAS-ViT-XS (2024) | 3.2 | 0.56 | 0.85 | 224 | 300 | 77.5 |
| EMO-5M (2023) | 5.1 | 0.90 | Failed | 224 | 300 | 78.4 |
| RepViT-M1.0 (2024) | 6.8 | 1.10 | 0.85 | 224 | 300 | 78.6 |
| **iFormer-S** | 6.5 | 1.09 | **0.85** | 224 | 300 | **78.8** |
| ShuffleNetV2 1.5× (2018) | 3.5 | 0.30 | 1.16 | 224 | 300 | 72.6 |
| EdgeViT-XXS (2022) | 4.1 | 0.60 | 1.41 | 224 | 300 | 74.4 |
| SHViT-S2 (2024) | 11.4 | 0.37 | 1.10 | 224 | 300 | 75.2 |
| EfficientMod-xxs (2024) | 4.7 | 0.60 | 1.29 | 224 | 300 | 76.0 |
| SBCFormer-S (2024) | 8.5 | 0.90 | 1.02 | 224 | 300 | 77.7 |
| MobileOne-S3 (2023b) | 10.1 | 1.90 | 1.16 | 224 | 300 | 78.1 |
| SwiftFormer-S (2023) | 6.1 | 1.00 | 1.12 | 224 | 300 | 78.5 |
| GhostNetV3 1.3x$^{\dagger}$ (2024) | 8.9 | 0.27 | 1.24 | 224 | 600 | 79.1 |
| EfficientNet-B1 (2019) | 7.8 | 0.70 | 1.29 | 240 | 350 | 79.1 |
| FastViT-T12 (2023a) | 6.8 | 1.40 | 1.12 | 256 | 300 | 79.1 |
| RepViT-M1.1 (2024) | 8.2 | 1.30 | 1.04 | 224 | 300 | 79.4 |
| RepNeXt-M3 (2024) | 7.8 | 1.30 | 1.04 | 224 | 300 | 79.4 |
| FastViT-S12 (2023a) | 8.8 | 1.80 | 1.26 | 256 | 300 | 79.8 |
| MNV4-Conv-M (2024) | 9.2 | 1.00 | 1.08 | 256 | 500 | 79.9 |
| **iFormer-M** | 8.9 | 1.64 | **1.10** | 224 | 300 | **80.4** |
| MobileViT-XXS (2021) | 1.3 | 0.40 | 2.12 | 256 | 300 | 69.0 |
| MobileViTV2-0.5 (2022) | 1.4 | 0.50 | 9.47 | 256 | 300 | 70.2 |
| ShuffleNet v2 2.0× (2018) | 7.4 | 0.59 | 1.94 | 224 | 300 | 74.9 |
| EdgeViT-XS (2022) | 6.7 | 1.10 | 1.79 | 224 | 300 | 77.5 |
| Mobile-Former-294M (2022b) | 11.4 | 0.29 | 2.66 | 224 | 450 | 77.9 |
| MobileViTV2-1.0 (2022) | 4.9 | 1.80 | Failed | 256 | 300 | 78.1 |
| EfficientMod-xs (2024) | 6.6 | 0.80 | 2.13 | 224 | 300 | 78.3 |
| MobileViT-S (2021) | 5.6 | 2.00 | 3.55 | 256 | 300 | 78.4 |
| CMT-Ti (2022) | 11.3 | 687 | Failed | 160 | 300 | 79.2 |
| Mobile-Former-508M (2022b) | 14 | 0.51 | 3.33 | 224 | 450 | 79.3 |
| SHViT-S4 (2024) | 16.5 | 0.99 | 1.48 | 224 | 300 | 79.4 |
| EfficientViT-B1-r224 (2023) | 9.1 | 0.52 | 2.38 | 224 | 350 | 79.4 |
| MobileOne-S4 (2023b) | 14.8 | 2.98 | 1.74 | 224 | 300 | 79.4 |
| LowFormer-B1 (2024) | 17.9 | 1.41 | 1.90 | 224 | 300 | 79.9 |
| SBCFormer-B (2024) | 13.8 | 1.60 | 1.44 | 224 | 300 | 80.0 |
| EfficientNet-B2 (2019) | 9.2 | 1.00 | 1.69 | 260 | 350 | 80.1 |
| CAS-ViT-S (2024) | 5.8 | 0.93 | 1.82 | 224 | 300 | 80.2 |
| GhostNetV3 1.6x$^{\dagger}$ (2024) | 12.3 | 0.40 | 1.49 | 224 | 600 | 80.4 |
| EfficientViT-B1-r288 (2023) | 9.1 | 0.86 | 3.87 | 288 | 450 | 80.4 |
| FastViT-SA12 (2023a) | 10.9 | 1.90 | 1.50 | 256 | 300 | 80.6 |
| MNV4-Hybrid-M (2024) | 10.5 | 1.20 | 1.75 | 256 | 500 | 80.7 |
| SwiftFormer-L1 (2023) | 12.1 | 1.60 | 1.60 | 224 | 300 | 80.9 |
| EfficientMod-s (2024) | 12.9 | 1.40 | 2.57 | 224 | 300 | 81.0 |
| SBCFormer-L (2024) | 18.5 | 2.70 | 1.89 | 224 | 300 | 81.1 |
| RepViT-M1.5 (2024) | 14.0 | 2.30 | 1.64 | 224 | 300 | 81.2 |
| LowFormer-B1.5 (2024) | 33.9 | 2.57 | 3.02 | 224 | 300 | 81.2 |
| RepNeXt-M4 (2024) | 13.3 | 2.30 | 1.47 | 224 | 300 | 81.2 |
| CAS-ViT-M (2024) | 12.4 | 1.89 | 2.46 | 224 | 300 | 81.4 |
| **iFormer-L** | 14.7 | 2.63 | **1.60** | 224 | 300 | **81.7** |