# OpenReview forum: "IFORMER: INTEGRATING CONVNET AND TRANSFORMER FOR MOBILE APPLICATION"
_ICLR.cc/2025/Conference — ICLR 2025 Poster_

### Official Review · Reviewer_aPLf · 2024-10-28

**Soundness:** 4
**Presentation:** 3
**Contribution:** 3
**Rating:** 8
**Confidence:** 3

**Summary:**

Summary

This paper proposes a mobile friendly vision network that improves the latency and accuracy by combining the strengths of both CNNs and ViTs. The novel aspect of this work is the single head modulation self-attention (SHMA). This SHMA learns spatial context through optimized self-attention. It takes ConvNext as the base model and improves it further with various techniques. The authors streamline the ConvNeXt architecture, making it suitable for real-time use on mobile devices, such as the iPhone 13, focusing on reducing latency rather than FLOPs or parameter count. The combined techniques lead to more than 80% top-1 accuracy with 1.1ms latency on iphone 13. Overall a great contribution to the research community.

**Strengths:**

1. fast local representation capacity of convolution and the efficient global modeling proficiency of the proposed SHMA
2. A series of novel techniques such as stack of overlapping convolution instead of aggressive non-overlapping patch in the early layers
3. The model is structured in four stages. The early stages use fast convolution to capture local features efficiently, using a modified and lightweight version of ConvNeXt optimized for mobile latency.
4. In the lower-resolution stages, self-attention is used to model long-range dependencies. To address the challenges of traditional multi-head self-attention (MHA), the authors propose SHMA, which uses a single-head attention mechanism to minimize memory costs while retaining high performance. SHMA reduces latency by optimizing reshaping operations and leveraging spatial context interactions. SHMA is combined with a parallel feature extraction branch to enhance feature representation. The outputs from both branches are fused to enable dynamic information exchange, mitigating any performance drop caused by simplifying MHA

**Weaknesses:**

1. What is the runtime complexity of iFormer network?
2. When running on iPhone (mobile device), what is the peak memory consumption?
3. How long the iPhone charge will last if an iFormer based app is run on certain fps?

**Questions:**

See weakness

---

> ### Author Response · Authors · 2024-11-18
> **Response to Reviewer aPLf**
>
> We thank the reviewer for the positive feedback!
>
> > Q1: What is the runtime complexity of iFormer network?
>
> Given an input $\ \mathbf{x}\in \mathbb{R}^{C\times H\times W}$ and a window size of P $\times$ P, as detailed in Section E, the computational complexity of iFormer is as follows:
> $\Omega (\text{SHMA}) = 4HWC^2 \text{(QKV and output projection)} +
> HWC \text{(element-wise product of modulation)} +
> 2P^2HWC \text{(self-attention)}$,
>
> $\Omega (\text{FFN}) = 8HWC^2.$
>
> In image classification, we do not utilize window attention since the feature size is 14$\times$ 14 in stage 3 (it equals to the window attention when P=14). In downstream tasks, we adopt a window size of P=16.
>
> > Q2: When running on iPhone (mobile device), what is the peak memory consumption?
>
> Currently, we employ the Xcode benchmark tool to evaluate the latency of our models. However, this tool does not support detailed memory consumption. We are actively exploring alternative methods to monitor the peak memory usage of iFormer on the iPhone.
>
> > Q3: How long the iPhone charge will last if an iFormer based app is run on certain fps?
>
> We appreciate your suggestions, and we plan to develop an app that provides more comprehensive information when running iFormer on the iPhone, including estimated battery life, peak memory usage, and other relevant metrics.

---

> > ### Comment · Reviewer_aPLf · 2024-11-23
> >
> > Thank you for you response. I will keep my score.

---

### Official Review · Reviewer_fhTG · 2024-10-31

**Soundness:** 2
**Presentation:** 2
**Contribution:** 2
**Rating:** 6
**Confidence:** 2

**Summary:**

This paper presents a new family of mobile hybrid vision networks, called iFormer,  by integrating the fast local representation capacity of convolution with the efficient global modeling ability of self-attention.

**Strengths:**

1. The paper is easy to follow, with clear writing and presentation.
2. Evaluation results are comprehensive.

**Weaknesses:**

1. How does this method compare with neural architecture search (NAS) methods?

2. How does the designed model perform on other mobile devices, such as NVIDIA Jetson Nano or Raspberry Pi?

**Questions:**

Please see the weaknesses.

---

> ### Author Response · Authors · 2024-11-18
> **Response to Reviewer fhTG**
>
> We would like to express our gratitude to the reviewer for their insightful comments and questions.
> > Q1: How does this method compare with neural architecture search (NAS) methods?
>
> In our study, we have compared two NAS methods, namely MNV4-Conv and EfficientFormerV2. iFormer demonstrated superior performance as depicted in Tables 3 and 4. NAS relies on the pre-defined search space of operators with the goal of identifying the optimal combinations and configurations of various operators. The innovative and efficient SHMA block significantly outperforms existing operators, enabling iFormer to achieve a better trade-off between accuracy and latency. As a general methodology, NAS can also be applied to iFormer and propel it towards a new SOTA.
>
> > Q2: How does the designed model perform on other mobile devices, such as NVIDIA Jetson Nano or Raspberry Pi?
>
> Due to the limitation of available devices and time constraints, we have not evaluated iFormer on a broader range of devices. In the future, we plan to expand the evaluation of iFormer to more platforms, including NVIDIA Jetson Nano and Android devices.
> It is important to note that designing a model performing well across multiple hardware platforms is challenging, as varying memory and computational architectures can lead to different behaviors.

---

> > ### Comment · Reviewer_fhTG · 2024-11-26
> >
> > Thank you for your response. I will maintain my score.

---

### Official Review · Reviewer_waKF · 2024-11-05

**Soundness:** 3
**Presentation:** 3
**Contribution:** 3
**Rating:** 6
**Confidence:** 5

**Summary:**

This paper presents a mobile hybrid vision network, iFormer. The paper goes from ConvNeXt to a lightweight mobile network. iFormer removes memory-intensive operations in MHA and employs an efficient modulation mechanism. The author conduct standard benchmark experiments on ImageNet, COCO and ADE20K.

**Strengths:**

I think the logic of exploration in this article, starting with ConvNeXt, first “lightening” the ConvNeXt to create a streamlined
lightweight network, then exploring the attention module, is reasonable.

I think the analysis about “cosine similarity between multiples” proves that using a single attention is good and worth supporting.

I think the experiment reported in this paper is comprehensive (imagenet, coco, ade-20k). The paper also reports some knowledge distillation results, which is suitable in mobile network papers.

**Weaknesses:**

1:
Single head self-attention has been conducted in "Shvit: Single-head vision transformer with memory efficient macro design" .

Alternative to standard self-attention has been conducted in GhostNetV2.

Modulation in the token mixer module has been conducted in Conv2Former.

This paper references many related methods, and while that is one approach, I don't think it stands out. Although such research is a decent format, I believe it impacts the novelty of this paper.

2:
The process of evolving from the ConvNeXt baseline to the lightweight iFormer may not apply to slightly larger models, and some steps show very minimal improvements, making them hard to justify.

**Questions:**

The largest model shown by iFormer, iFormer-L, is only about 15M, which isn’t considered large, even for edge devices, especially since recent edge LLMs can reach 1B parameters. I wonder how well a larger iFormer (around 100M) would perform.

---

> ### Author Response · Authors · 2024-11-18
> **Response to Reviewer waKF (Part 1/2)**
>
> W1:
>
> We would like to clarify that iFormer is not a simple concatenation of multiple existing methods. Instead, we propose several novel designs aiming at lightening ConvNeXt.
> > Single head self-attention has been conducted in "Shvit: Single-head vision transformer with memory efficient macro design" .
>
> 1.  Single Head Attention (SHA) differs from its counterpart in SHViT in the following aspects: **In terms of motivation**, iFormer explores efficient attention mechanisms specifically tailored for the on-device environment, whereas SHViT is geared towards general-purpose GPUs, which may exhibit different hardware characteristics.
> **In terms of methodology**, we utilize single head attention with more channels, while SHViT employs fewer than 1/4 of channels for attention. The reduced number of channels can result in a lower rank of the attention matrix, potentially degrading its expressiveness. Additionally, the split and concatenate operations in SHViT introduce extra runtime.
>
> The experimental results in Table 15 show that iFormer achieves a better trade-off than SHViT.
>
> We also conduct a more fair comparison with SHViT. We use the SHA baseline in Table 1, specifically denoted as 'SHA' in Figure 2. Then we move toward SHViT as follows:
>
> | Model                       | Params | GMACs | Latency | Top-1 |
> |:----------------------------|:------:|------:|--------:|------:|
> | SHA Baseline without Modulation  |  9.9M  | 1758M |  1.12ms |  79.8 |
> | + split                     |  9.9M  | 1758M |  1.18ms |     - |
> | + attention on 1/4 channels |  8.3M  | 1547M |  1.02ms |     - |
> | + concat                    |  8.7M  | 1579M |  1.11ms |  79.5 |
>
> The discussion can also be found in the Section D of the revised version (highlighted in blue). It can be observed that split and concat operations introduce additional runtime. Moreover, the SHA of iFormer demonstrates stronger performance than the counterpart of SHViT under similar latency (79.8 v.s. 79.5). This improved performance may be attributed to the greater number of channels in the attention mechanism
> > Alternative to standard self-attention has been conducted in GhostNetV2.
>
> 2. SHMA is fundamentally different from the attention mechanism in GhostNetV2, which decomposes spatial interactions into horizontal and vertical interactions and behaves more like the Mlp-mixer [1]. In contrast, SHMA integrates the single head self-attention with modulation. Besides, the experiments in Table 3 suggest that iFormer demonstrates a superior and more efficient alternative to self-attention than GhostNetV2 (with V3 being the upgraded version).
>
> > Modulation in the token mixer module has been conducted in Conv2Former.
>
> 3. Thanks for bringing Conv2Former to our attention, we include discussions about it in the related work in the revision. Modulation is a general methodology for enhancing dynamic interactions. SHMA distinguishes itself as a novel approach from existing modulation techniques including Conv2Former, VAN, and FocalNet in the direct integration of self-attention into the context branch of modulation. This integration enables SHMA to capture more informative context.
>
> [1] Mlp-mixer: An all-mlp architecture for vision. NeurIPS 2021.

---

> ### Author Response · Authors · 2024-11-18
> **Response to Reviewer waKF (Part 2/2)**
>
> W2 and Q:
>
> Although iFormer is designed for mobile-device applications, the combination of fast local representation capacity of convolution and the efficient global modeling proficiency of the proposed SHMA enables its scalability for a broader range of applications.
> To demonstrate the scalability of iFormer, we developed a larger model named iFormer-H with 99M parameters and trained it for 300 epochs following the same strategy outlined in Section B of the revision (highlighted in blue). It is important to note that we add drop path and layer scale, which are commonly used in the training of larger models [1,2,3]. The performance results are provided as follows:
>
> | Model             | Params | GMACs | Top-1 |
> |:------------------|:------:|------:|------:|
> | ConvNeXt-Base[1]  |  89M   | 15.4G |  83.8 |
> | TransNeXt-Base[2] |  90M   | 18.4G |  84.8 |
> | iFormer-H         |  99M   | 15.5G |  84.8 |
> | MaxViT-Base [3]   |  120M  | 24.0G |  84.9 |
>
> A highlight from the results is that iFormer is not specifically designed or trained for this scale. Despite this, iFormer-H outperforms ConvNeXt, achieving a 1.0% increase in accuracy while maintaining a similar number of FLOPs. Additionally, it demonstrates comparable performance to TransNeXt-Base, despite utilizing fewer FLOPs. These findings indicate the potential for broader applications of iFormer. We will also open source this model with source code in the Github and plan to explore larger models suitable for mobile devices in future work.
>
> [1] A ConvNet for the 2020s. CVPR 2022
>
> [2] TransNeXt: Robust Foveal Visual Perception for Vision Transformers. CVPR 2024
>
> [3] MaxViT: Multi-Axis Vision Transformer. ECCV 2022

---

> > ### Comment · Reviewer_waKF · 2024-11-26
> > **Nice Response**
> >
> > Thank you for your detailed response. It has clarified some of my questions, and I will increase my score from 5 to 6.

---

### Official Review · Reviewer_XLkf · 2024-11-06

**Soundness:** 3
**Presentation:** 3
**Contribution:** 3
**Rating:** 6
**Confidence:** 4

**Summary:**

This paper introduces a new family of mobile hybrid vision networks. By integrating the rapid local representation capability of convolution with the efficient global modeling ability of self-attention, the proposed architecture, iFormer, achieves significant performance in classification and several downstream tasks, while maintaining low latency on mobile devices for high-resolution inputs.

**Strengths:**

1. The study of model architecture could inspire further exploration in designing more efficient architectures.
2. The paper is well-organized and easy to follow.

**Weaknesses:**

1. In Table 1, iFormer-S achieves the same latency as RepViT-M1.0 with slightly fewer parameters, yet in larger variants, iFormer achieves lower latency with substantially more parameters compared to RepViT. What is the reason for this difference?
2. Some studies are not included in the comparison or the related wotk section, such as [1, 2].

[1] Cmt: Convolutional neural networks meet vision transformers.

[2] Learning efficient vision transformers via fine-grained manifold distillation.

**Questions:**

please refer to weakness

---

> ### Author Response · Authors · 2024-11-18
> **Response to Reviewer XLkf**
>
> We appreciate the reviewer's insightful comments and questions!
>
> Q1:
>
> We apologize for the mistake. The latency of RepViT-M1.0 is 1.54 ms, rather than 1.64 ms. We have updated this information in Tables 3 and 4 in the revision (highlighted in blue). Additionally, we have conducted a thorough review of all experimental results included in the paper.
>
> Q2:
>
> We have included these two papers in the related work section and included a comparison of CMT in Table 15. These updates are highlighted in blue in the Section 2.2 Efficient Vision Transformers of the revision.

---

> > ### Comment · Reviewer_XLkf · 2024-11-27
> >
> > Thanks you for the response. All of my concerns have been addressed. I will keep my initial score.

---

### Official Review · Reviewer_wKqn · 2024-11-06

**Soundness:** 3
**Presentation:** 3
**Contribution:** 3
**Rating:** 6
**Confidence:** 4

**Summary:**

This paper designs iFormer, a new family of efficient mobile vision networks combining ConvNet and Transformers. The iFormer evolves from ConvNeXt with a series of efficiency designs.

Single-Head Modulated Attention(SHMA) is proposed as substitutional Transformer blocks to replace part of the Conv blocks in later stages of the enhanced ConvNeXt. SHMA replaces multi-head attention with single-head attention to improve efficiency and introduces a modulation mechanism to boost performance.

The resulting iFormer series achieves the best performance compared with state-of-the-art mobile-level models on different downstream tasks with lower latency.

**Strengths:**

This paper is well-organized and easy to follow. Detailed design specifications and comprehensive experiments enhanced the integrity of the article and demonstrated its contributions.

The main contribution, SHMA, provides a new approach to designing efficient attention and Transformer blocks. The resulting iFormer series outperforms sota baseline mobile networks with stronger performance and lower latency.

**Weaknesses:**

W1:

The motivation and necessity of substituting half of the conv blocks at the third stage and all blocks at the last stage into Transformer blocks in ConvNeXt are still not very clear. From Figure 2, changing the conv blocks into SHA blocks gains a 0.4% improvement in performance but is also 0.12 ms (about 10%) slower. I'd like to know further explanation for this design and ablation studies on the choice of stages or different ratios of Conv versus Transformer blocks if possible.


W2:

According to the citation of SHViT in this paper, I suppose the SHA refers to the Single-head self-Attention in SHViT design. But in Figure 4, full channels of input (CxHxW) are projected to Q/K/V (CxL) which does not align with the design of SHA in SHViT but looks like the traditional definition of Single-head attention that performs a self-attention on all channels of input using a single head.

Considering there are limited words about the details of SHA in this paper, I would expect further specification of which SHA is used in iFormer and comply with the pipeline figure accordingly.

W3:

In this paper, the additional reshaping operations in MHA are considered as the reason for the slower inference speed compared with SHA. But multiple factors have an impact on the runtime speed difference and there's no evidence to support the extra runtime only or mainly comes from extra reshapings.

First, depending on the code implementation, replacing MHA with Single-head self-Attention may remove the reshaping operation in self-attention, but also introduce additional split and concat operations. And generally, split and concat operations cost more memory and are slower than reshape.

Secondly, SHA applies self-attention on fewer channels, which largely reduces the computational cost and speeds up runtime.

Therefore I suggest the authors conduct an ablation study or provide empirical evidence to isolate the impact of reshaping operations versus other factors like split/concat operation and reduced self-attention channels on the inference speed. This would help clarify the main factors contributing to SHA's efficiency and provide a more comprehensive understanding of the proposed method.

**Questions:**

1. What is the motivation and justification for the necessity of the design that replaces half of the third stage and full last stage conv blocks with transformer blocks? Please refer to Weaknesses 1.

2. I wonder what is the performance and latency of MHA as the missing step between 'kernel sz.' and 'SHA' in Figure 2.

---

> ### Author Response · Authors · 2024-11-18
> **Response to Reviewer wKqn (part 1/2)**
>
> We thank the reviewer for the thorough and constructive feedback.
>
> W1 and Q1:
>
> We would like to clarify that SHA serves as an intermediate design towards SHMA. So changing the convolutional blocks yields a 1.0% improvement.
> > I'd like to know further explanation for this design and ablation studies ... if possible.
>
> We conduct ablation studies on the choice of Conv versus ViT blocks, which ultimately lead to the architecture of iFormer. We choose the model after enlarging the kernel size as a start point, then we progressively replace the convolutional blocks in Stages 3 and 4. We do not modify Stages 1 and 2 as they have larger spatial dimensions, which significantly increase memory requirements for the self-attention mechanism.
> We present these findings here and have incorporated them into the Section 5 (Choice of Conv v.s. ViT Blcoks) of the revised manuscript, highlighted in blue.
>
> | Model                                                            | Params | GMACs | Latency | Top-1 |
> |:-----------------------------------------------------------------|:------:|------:|--------:|------:|
> | Baseline                                                         |  9.4M  | 1760M |   1.00ms |  79.4 |
> | Replacing 22% Conv Blocks in Stage 3 as SHA                      |  9.1M  | 1724M |  1.02ms |  79.5 |
> | Replacing 22% Conv Blocks in Stage 3 as SHMA                     |  9.2M  | 1739M |  1.04ms |  79.6 |
> | Replacing 50% Conv Blocks in Stage 3 as SHA                      |  8.8M  | 1689M |  1.04ms |  79.5 |
> | Replacing 50% Conv Blocks in Stage 3 as SHMA                     |  8.9M  | 1712M |  1.07ms |  79.8 |
> | Replacing 78% Conv Blocks in Stage 3 as SHA                      |  8.3M  | 1635M |  1.12ms |  79.3 |
> | Replacing 78% Conv Blocks in Stage 3 as SHMA                     |  8.5M  | 1685M |  1.17ms |  79.6 |
> | Replacing 100% Conv Blocks in Stage 3 as SHA                     |  7.9M  | 1599M |  1.17ms |  78.1 |
> | Replacing 100% Conv Blocks in Stage 3 as SHMA                    |  8.3M  | 1665M |  1.25ms |  79.0 |
> | Replacing 100% Conv Blocks in Stage 3 as SHMA and 100% in Stage 4 | 10.0M  | 1792M |  1.15ms |  80.4 |
>
> Since Stage 4 contains only two blocks, we do not conduct further split for the ratio. As shown in the above table, the ViT block incurs more runtime. By replacing half of the convolutional blocks in the third stage and all blocks in the final stage, we achieve a pretty trade-off between accuracy and latency.

---

> ### Author Response · Authors · 2024-11-18
> **Response to Reviewer wKqn (part 2/2)**
>
> W2, W3:
>
> We are sorry for any confusion caused by the unclear statement. We would like to clarify that our SHA has no relationship to SHViT. We have included more details about SHA in the Section D of the revised version (highlighted in blue). Additionally, we will release the source code in the final version.
> > According to the citation of SHViT in this paper, I suppose ... in SHViT design.
>
> The SHA in iFormer is different from SHViT in the following three key aspects. First, in terms of motivation, iFormer explores efficient attention mechanisms in an on-device environment, while SHViT focuses on general-purpose GPUs, which may exhibit distinct characteristics.
> > full channels of input (CxHxW) are ... single head.
>
> Second, in terms of methodology, the full channels of the input (CxHxW) are projected to Q/K (C/R X L) and V (C x L), where R is the head reduction ratio and is set to 2 in our models. We have updated the Figure 4 of the revision to reflect this. In contrast, SHViT splits the input and utilizes fewer than 1/4 of the channels for attention. This reduction in channels can lower the rank of the attention matrix, thereby degrading its representational capacity. Furthermore, we do not use split and concatenation operations as they tend to exacerbate memory access costs.
> > I suggest the authors conduct an ablation study ... the proposed method.
>
> To show this, we refer to the SHA baseline in Table 1, i,e., 'SHA' in Figure 2. Subsequently, we transit it toward SHViT as follows:
>
> | Model                       | Params | GMACs | Latency | Top-1 |
> |:----------------------------|:------:|------:|--------:|------:|
> | SHA Baseline without Modulation   |  9.9M  | 1758M |  1.12ms |  79.8 |
> | + split                     |  9.9M  | 1758M |  1.18ms |     - |
> | + attention on 1/4 channels |  8.3M  | 1547M |  1.02ms |     - |
> | + concat                    |  8.7M  | 1579M |  1.11ms |  79.5 |
>
> It can be observed that split and concat operations introduce additional runtime. Furthermore, the performance of the SHA in the SHViT exhibits a decline compared to its counterpart in iFormer under similar latency conditions (79.8 v.s. 79.5). This degraded performance may be attributed to the reduced number of channels in the attention mechanism. We have updated this information in the Section D of revision.
>
> Q2:
>
> Since we do not apply SHViT, the only difference between the MHA Baseline and the SHA Baseline in Table 1 is the number of heads, which pertains to reshaping. The performance and latency of the MHA serve as the missing step between 'kernel sz.' and 'SHA', represented by the MHA Baseline.

---

### Author Response · Authors · 2024-11-18
**Global Response**

We would like to express our heartfelt gratitude to all PCs, SACs, and Reviewers for their efforts.

And we would like to highlight our contributions as follows:
* The detailed exploration roadmap could inspire further exploration in designing  efficient architectures.
* Our newly introduced SHMA effectively minimizes memory costs while maintaining high performance.
* iFormer outperforms SOTA baselines across a comprehensive range of tasks.

As suggested by the reviewers, we have made the following revisions to the manuscript (including Supplementary Material):
* A more comprehensive discussion in the related work section.
* A more detailed illustration of SHMA.
* Corrections to writing errors related to RepViT and revision throughout the manuscript.
* Update the Top-1 accuracy of iFormer-L from 81.7 to 81.9. This improvement was achieved by employing a drop path rate of 0.1, without making any changes to other parameters.
* Additional ablation studies on the choice of convolutional blocks versus vit blocks.
* Further ablation studies examining scalability.
* An extended discussion on future work.
* Additional comparison including ablation studies and illustrations in relation to SHViT.
* Additional discussion of computation complexity.

We have highlighted all the modifications in blue, which will be removed following the rebuttal process.

---

### Author Response · Authors · 2024-11-22
**Dear Reviewer**

As the discussion period is approaching to an end, if our responses have not sufficiently addressed your concerns or if additional clarification is required, please let us know. We sincerely appreciate your dedication to reviewing our work, and we are grateful for your insightful comments and the considerable time you have invested in evaluating our paper.

---

### Meta-Review · Area_Chair_7WTc · 2024-12-15

**Metareview:**

This paper introduces iFormer that combines convolutional neural networks with Transformer-based architectures. Evolving from ConvNeXt, the proposed model integrates Single-Head Modulated Attention to replace parts of the Conv blocks in later stages, aiming to balance efficiency and performance. After rebuttal, all reviewers agree that this paper offers a well-structured and experimentally validated approach to mobile hybrid vision networks. The authors are encouraged to incorporate these updates into their final submission to strengthen the paper’s clarity, justification of design choices, and overall impact.

**Additional Comments On Reviewer Discussion:**

During the discussion stage, 4 out of 5 reviewers responded to the author's rebuttal and agreed that it addressed their main concerns. Given the positive scores and overall feedback, I recommend accepting this paper as a poster.

---

### Decision · Program_Chairs · 2025-01-22

Accept (Poster)